# Factors influencing weight management behavior among college students: An application of the Health Belief Model

**Maryam Saghafi-Asl¹, Soghra Aliasgharzadeh❶²\*, Mohammad Asghari-Jafarabadi³**

**1** Nutrition Research Center, School of Nutrition and Food Sciences, Tabriz University of Medical Sciences, Tabriz, Iran, **2** Student Research Committee, School of Nutrition and Food Sciences, Tabriz University of Medical Sciences, Tabriz, Iran, **3** Road Traffic Injury Research Center, Tabriz University of Medical Sciences, Tabriz, Iran

\* Soghra.aliasgharzadeh@gmail.com

**Data Availability Statement:** All relevant data are within the manuscript and its Supporting Information files.

## Abstract

### Background

Overweight and obesity have become a significant public health concern in both developing and developed countries. Due to the health implications of weight-reduction behaviors, it is important to explore the factors that predict their occurrence. Therefore, the present study was performed to examine factors affecting the behavioral intention of weight management as well as assess the predictive power of the Health Belief Model (HBM) for body mass index (BMI).

### Methods

This cross-sectional study was conducted among 336 female students recruited from dormitories of Tabriz University of Medical Sciences, using quota sampling technique. Data were collected by a structured questionnaire in seven parts (including perceived severity, perceived susceptibility, perceived benefit, perceived barrier, cue to action, self-efficacy in dieting and physical activity, and behavioral intention of weight management), based on the HBM. Structural equation modeling (SEM) was conducted to identify the relationship between HBM constructs and behavioral intention of weight management. Linear regression model was performed to test the ability of the HBM to predict students' BMIs.

### Results

Higher level of perceived threats (sum of perceived susceptibility and severity) ($\beta = 0.41$, $P<0.001$), perceived benefits ($\beta = 0.19$, $P = 0.009$), self-efficacy in exercise ($\beta = 0.17$, $P = 0.001$), and self-efficacy in dieting ($\beta = 0.16$, $P = 0.025$) scales was significantly related to greater behavioral intention of weight management. Moreover, perceived threat mediated the relationships between perceived cue to action, perceived benefits, self-efficacy in exercise, and weight management practices. The fit indices of the SEM model seemed acceptable. The final regression model explained approximately 40% of variance in BMI

**Funding:** This work was supported by the Tabriz University of Medical Sciences to MS-A. The funder had no role in study design, data collection and analysis, decision to publish, or preparation of the manuscript.

**Competing interests:** The authors have declared that no competing interests exist.

(P<0.001). Additionally, perceived severity, barrier, and self-efficacy in dietary life were the significant variables to predict students' BMIs.

## Conclusions

These findings suggest that health education programs based on the HBM needs to be integrated in preventive health programs and health interventions strategies to ensure adherence and well-being of the participants.

## Introduction

Overweight and obesity have become epidemic rising trends in both developed and developing countries [1–4]. According to estimates by World Health Organization (WHO) in 2016, there were approximately 1.9 billion overweight adults aged 18 years and above from which at least 650 million were obese [5]. The growing trend in the transition from overweight status to obesity often occurs at ages 18–29 years. Obesity is an important concerns of health care professionals, as it is accompanied by numerous physical and psychological problems including coronary heart disease, diabetes, and several cancers [6–8]. Obesity also imposes enormous financial burdens on both governments and individuals [9]. Several factors contribute to obesity including genetics and behavioral and environmental parameters such as physical activity and dietary behavior [10].

The collegiate period is a critical time for altering physical activity and dietary patterns which lead to weight gain of students [11, 12]. Thus, weight management remains an important health challenge for this population. Several preventive and treatment programs are applied for weight control [13]. However, compliance with weight-loss treatments varies among women for a range of reasons [13, 14]. Previous studies have shown that psychosocial factors such as perceptions about health and obesity, and self-efficacy play important roles in the success of weight loss and maintenance programs [15–17].

To develop effective weight management interventions for college students, it is important to understand the factors that predict the occurrence of appropriate weight reduction behavior. The Health Belief Model (HBM) is a health-specific social cognitive model that attempts to predict and explain why individuals change or maintain specific health behaviors [18]. This model assumes that individual involvement in health-related behaviors is determined by understanding six following constructs: *Perceived severity* (an individual's perception of the seriousness and potential consequences of the condition), *Perceived susceptibility* (an individual's assessment of their risk of getting a disease or condition), *Perceived benefit* (an individual's beliefs about whether the recommended behavior will reduce the risk or severity of impact), *Perceived barrier* (an individual's assessment of the difficulties and cost of adopting behaviors), *Cue to action* (the internal or external motivations promoting the desired behavior), and *Self-efficacy* (an individual's belief about their capabilities to successfully perform a new health behavior). These six constructs provide a conceptual framework for designing both long and short-term health behavior interventions (Fig 1) [18, 19].

Several studies examined the factors affecting weight control intention through HBM [20–23]. Park *et al.* examined factors affecting behavior intention of weight reduction among female middle-school students, using HBM [20]. They found that perceived threat (a sum of severity and susceptibility), cues to action, and perceived self-efficacy were significantly associated with behavioral intention of weight reduction for all respondents [20]. McArthur et al.

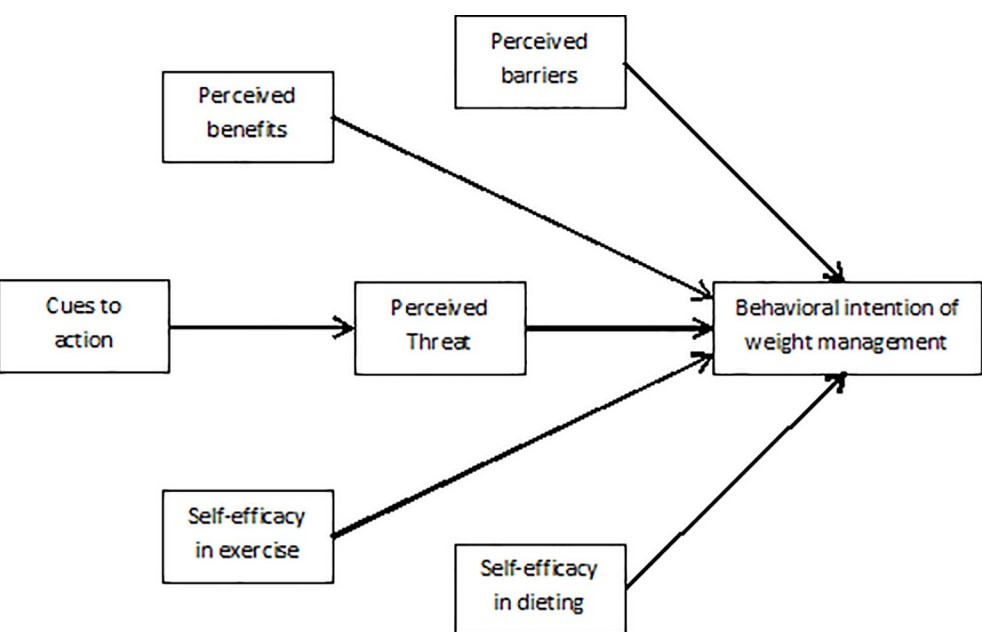

**Fig 1. Theoretical framework of Health Belief Model applied to behavioral intention of weight management.**

tested the predictive power of HBM (which consisted of perceived susceptibility, perceived severity, perceived benefits, perceived barriers, and cues to action) for body mass index (BMI) among a college student sample [21]. They found significant positive associations between ratings on the perceived susceptibility, perceived barriers, and perceived benefits scales and BMI. Findings also revealed significant inverse associations between ratings on the perceived severity, and external cues to action scales and BMI [21].

To the best of our knowledge, no research has been conducted on the whole HBM constructs for the prediction of weight management among college students. Therefore, the present study aimed to (1) develop and assess the validity and reliability of an HBM-based questionnaire for weight management behavior, (2) explore the effects of all HBM constructs on weight management behaviors among college students. Based on the second objective, we proposed the following hypotheses:

H1: Behavioral intention of weight management will be positively influenced by perceived threat, perceived benefits, and self-efficacy in dieting and exercise. H2: Perceived barriers will negatively influence behavioral intention of weight management. H3: Perceived threat will mediate relationship between cues to action and behavioral intention of weight management, and (3) determine the predictive power of HBM constructs for the BMI of students.

## Methods

### Research design and sampling

This cross-sectional study was conducted among Iranian students from dormitories of Tabriz University of Medical Sciences from June to September 2018. It is suggested that the ratio between the sample size and the number of model parameters in the range of 10:1 or even 20:1 seem appropriate [24]. The hypothesised model in this study incorporated 22 parameters. Considering a 16:1 ratio, the sample size was determined to be 352 for the study. In order to allow for potential missing data, the initial sample size was set at 380. In the process of sampling, a sample of 380 subjects who agreed to participate was evaluated, 14 of whom given

imperfect data in questionnaire were excluded from the study. Therefore, the final sample size in analysis was 366. The subjects were selected through quota sampling method; all dormitories were chosen then in proportion of number of students' resident in each dormitory and in accordance with the estimated sample size, a quota was assigned to each one and the convenience sampling from these dormitories was carried out. Data were collected through personal interviews, using a structured questionnaire. Informed consent was obtained from all participants, before the onset of the study.

## Measurement tool

The first version of the questionnaire used in measuring HBM variables was derived from Park (2011) and McArthur *et al.* (2017) [20, 21]. Eighty-nine statements were included and represented 8 perceptual and behavioral categories, as follow: 13 questions on perceived severity consisting of 3 subscales (emotional/mental, health, physical health/ fitness, and social professional); 7 questions on perceived susceptibility consisting of 2 subscales (lifestyle and environmental); 14 questions on perceived barriers consisting of 3 subscales (practical concerns, emotional/ mental health, and awareness); 13 questions on perceived benefits including 3 subscales (emotional/ mental health, physical health/ fitness, and social/ professional); 12 questions on cues to action consisting of 2 subscales (internal and external cues to action); 18 questions on self-efficacy in dieting including 2 subscales (Habits and preferences and Emotional/mental health); 7 questions on self-efficacy in exercise, and 5 questions on behavioral intention of weight management consisting of 2 subscales (dieting and exercising). All statements were rated using a five-point Likert scale ranging from 1 (strongly disagree) to 5 (strongly agree). In order to determine the content validity, ten specialists and professionals (outside the team) in the field of Health and Nutrition were consulted. Then, based on the Lawshe's Table, items with higher values of Content Validity Ratio (CVR) (i.e. higher than 0.62 for 10 people) and Content Validity Index (CVI) (*i.e.* higher than 0.75) were considered acceptable [25]. CVI and CVR showed satisfactory results for each item (CVI range: 0.78–1.00 and CVR range: 0.80–1.00). Reliability was calculated using internal consistency (Cronbach's Alpha). Alpha coefficients equal to or higher than 0.70 were considered satisfactory [26]. The overall reliability of the instrument based on the Cronbach's alpha, was 0.92. To assess the test-retest reliability of the questionnaire, a subgroup of 30 randomly selected students were asked to repeat the survey after a two-week interval. Intraclass correlation coefficient (ICC) was computed to evaluate the stability over time. ICC indicated excellent agreement (ICC = 0.86).

## Statistical analysis

Data analyses were conducted using STATA version 12. The characteristics and beliefs of the participants were described, using means (SD) and frequencies (percentages), wherever appropriate. Weight groups were divided into three categories: underweight (BMI<18.5 kg/m$^2$), normal weight (18.5$\leq$BMI<25 kg/m$^2$), and overweight (BMI$\geq$25 kg/m2). There were few obese students, who were put into the overweight group. Chi-square tests were applied to analyze categorized variables. The mean differences were determined by Kruskal Wallis test. In the case of significant results, Mann-Whitney U test with Bonferroni correction was used to assess the pair-wise comparisons.

Multiple imputation in expectation–maximization (EM) algorithm method was run to manage missing data [27]. Path analysis was used as a tool for structural equation modeling (SEM) to determine the relationship between HBM constructs and behavioral intention of weight management and recognize direct and indirect influence of independent variables toward dependent variables. The magnitude of the relationship was measured by path

coefficients and correlations, as standardized estimates. Goodness of fit indices selected for model evaluation were: normed chi-square ($\chi 2/df$, values lower than 5 were accepted); comparative fit index (CFI, values greater than 0.90 were accepted); Tacker Lewis index (TLI, values greater than 0.90 were accepted); standardized root mean squared residual (SRMR, values lower than 0.05 were accepted); and root mean square error of approximation (RMSEA, values lower than 0.08 were accepted) [28, 29].

A hierarchical linear regression analysis was performed to estimate the relationships between HBM scales, demographic characteristic, and BMI. P-Values less than 0.05 were considered as statistically significant.

# Results

## Baseline characteristics

A total of 336 students completed the questionnaires. The mean age of the students was 22.02 (±3.02; range, 18–43) years. Based on self-reported weight and height data, the mean BMI was 22.62 (±3.17; range, 15.63–32.72) $kg/m^2$. The baseline characteristics of the participants based on three weight groups are presented in Table 1. The marital status of the students was significantly different among weight groups (P = 0.002). The majority (89.9%) of the students were single.

There was a significant relationship between family history of obesity and weight status of the student (P = 0.004). Approximately, 68 percent of the participants had at least one obese family member. Nearly half of the students had experience trying to lose weight. This experience differed significantly among weight groups (P<0.001). Most of the students controlled their diet and exercised to lose their weight. More than half of the students responded that they attempted to manage their weight to improve their appearance, while about one-thirds did so for health. There were significant differences in "the reasons for weight reduction" among under- and normal-weight and overweight groups (P<0.001). The socioeconomic status of the students was not significantly different among weight groups.

## Weight-related beliefs of the participants by weight status

Weight-related beliefs of the students comprising the mean scales and related subscales ratings (SD), and the Cronbach's alpha are presented in Table 2. The mean scores of the 13-item perceived severity of the overweight consequences were 3.26±0.76 for all respondents that showed significant differences among the three groups (P≤0.001). Students in the underweight group showed the highest mean score for perceived severity (3.84±0.57). The beliefs for the physical health/fitness subscale received higher ratings than the other severity subscales (3.44±0.85). Underweight and normal weight students rated the emotional/mental health subscale higher than overweight students (P≤0.001). The mean score of physical health/fitness and social/professional subscales showed significant differences among the three groups (P≤0.001).

The mean score of the total perceived susceptibility of obesity risk was 3.46±0.76 for all respondents. Students in the underweight group had the highest score (3.64±0.66); however, there were no significant differences among the three groups.

The mean score of the 14-item perceived barriers to adopting healthy eating and physical activity habits were 2.94±0.75 for all respondents that showed significant differences among the three groups (P≤0.001). In addition, students in overweight group showed the strongest perceived barrier (3.60±0.73); followed by students in the normal weight (2.81±0.64), and underweight (2.39±0.59) group. Beliefs from the emotional/mental health subscale received higher rating than other ones.

**Table 1. Baseline characteristics of the study participants.**

| Variable | All (n = 336) | Underweight (n = 28) | Normal weight (n = 236) | Overweight (n = 72) | P-value |
|---|---|---|---|---|---|
| *Marital status* | | | | | |
| Single | 302(89.9) | 28(9.27)[a] | 217(71.85)[a] | 57(18.87)[b] | 0.002** |
| Married | 34(10.1) | 0(0.00)[a] | 19(55.88)[a] | 15(44.12)[b] | |
| *Education level* | | | | | |
| BSc degree | 220(65.48) | 19(8.64)[a] | 155(70.45)[a] | 46(20.91)[a] | 0.670* |
| MSc degree | 36(10.72) | 1(2.78)[a] | 24(66.67)[a] | 11(30.55)[a] | |
| Ph.D. degree | 80(23.81) | 8(10.00)[a] | 57(71.25)[a] | 15(18.75)[a] | |
| *Obese family member* | | | | | |
| Yes | 229(68.15) | 14(6.11)[a] | 158(69.00)[b] | 57(24.89)[c] | 0.004* |
| No | 107(31.85) | 16(14.95)[a] | 78(72.90)[b] | 13 (12.15)[c] | |
| *Experience in weight loss behavior* | | | | | |
| Yes | 146(43.45) | 1(0.68)[a] | 95(60.07)[b] | 50(34.25)[c] | <0.001* |
| No | 190(56.55) | 27(14.21)[a] | 141(74.21)[b] | 22(11.58)[c] | |
| *Experience of diet therapy* | | | | | |
| Yes | 91(27.08) | 1(1.10)[a] | 50(54.94)[b] | 40(43.95)[c] | <0.001* |
| No | 245(72.92) | 27(11.02)[a] | 186(75.92)[b] | 32(13.06)[c] | |
| *Experience of exercise therapy* | | | | | |
| Yes | 156(46.43) | 4(2.56)[a] | 97(62.18)[b] | 55(35.26)[c] | 0.001* |
| No | 180(53.57) | 24(13.33)[a] | 139(77.22)[b] | 17(9.44)[c] | |
| *Experience of medical treatment* | | | | | |
| Yes | 20(5.95) | 1(5.00)[ab] | 8(40.00)[a] | 11(55.00)[b] | 0.004* |
| No | 316 (94.05) | 27(8.55)[ab] | 228(72.15)[a] | 61(19.30)[b] | |
| *Reason of weight management behavior* | | | | | |
| Health | 80(29.73) | 3(3.75)[a] | 51(63.75)[a] | 26(32.50)[a] | <0.001* |
| Better appearance | 147(54.65) | 8(5.44)[a] | 116(78.91)[a] | 23(15.65)[b] | |
| Health and better appearance | 38(14.13) | 0(0.00)[a] | 16(42.11)[a] | 22(57.89)[b] | |
| Other | 4(1.49) | 0(0.00)[a] | 4(100.00)[a] | 0(0.00)[a] | |
| *Socioeconomic status* | | | | | |
| Low | 28(8.33) | 1(3.57)[a] | 22(78.57)[a] | 5(17.86)[a] | 0.064* |
| Middle | 208(61.94) | 19(9.14)[a] | 146(70.19)[a] | 43(20.67)[a] | |
| High | 100(29.76) | 8(8.00)[a] | 68(68.00)[a] | 24(24.00)[a] | |

Data are expressed as frequency (percent)

*P value based on Chi-squared test.

**P value base on Fisher's Exact test.

[a, b, c] Values within a row with the same letter indicate no significant difference. Any difference between two values carrying different letters is significant at 0.05 based on Mann—Whitney U with Bonferroni Correction.

The mean score of the 13-item perceived benefits to adopting healthy eating and physical activity habits were 3.73±0.67 for all respondents. There were no significant differences in mean rating on total scale among the three groups. The Emotional/mental health subscale construct received higher rating than other ones.

The mean score of the perceived cues to action for weight management was 3.49±0.70 for all respondents. Normal-weight students had the highest score (3.54±0.65), but there were no significant differences among the three groups. The mean rating of external and internal cues to action were not different among the study groups.

**Table 2. The students' average score of weight-related beliefs.**

| | All | Underweight | Normal weight | Over weight | P-value |
|---|---|---|---|---|---|
| **Perceived Severity** | | | | | |
| Emotional/mental health subscale (Cronbach α = 0.89) | 3.41±0.96 | 3.77±0.85[a] | 3.50 ±0 .88[a] | 2.97±1.08[b] | ≤0.001 |
| Physical health/fitness subscale (Cronbach α = 0 .84) | 3.44±0.85 | 4.00±0.55[a] | 3.52 ± 0 .77[b] | 2.93±1.95[c] | ≤0.001 |
| Social/professional subscale (Cronbach α = 0.71) | 2.90±0.89 | 3.70±0.62[a] | 2.93 ± 0.86[b] | 2.48 ±0.80[c] | ≤0.001 |
| Total (Cronbach α = 0.90) | 3.26±0.76 | 3.84 ±0.57[a] | 3.33 ± 0.69[b] | 2.80±0.81[c] | ≤0.001 |
| **Perceived Susceptibility** | | | | | |
| Lifestyle subscale (Cronbach α = 0.82) | 3.50±0.83 | 3.69 ± 0.68 | 3.53 ±0.80 | 3.27 ± 0.93 | 0.051 |
| Environmental subscale (Cronbach α = 0.72) | 3.37±0.90 | 3.51 ± 0.80 | 3.40 ±0.85 | 3.22 ±1.03 | 0.286 |
| Total (Cronbach α = 0.84) | 3.46±0.76 | 3.64 ± 0.66 | 3.50 ± 0.71 | 3.26 ±0.88 | 0.075 |
| **Perceived Barriers** | | | | | |
| Practical concerns subscale (Cronbach α = 0.78) | 2.91±0.82 | 2.41 ±0.69[a] | 2.80 ± 0.74[b] | 3.50 ±0.85[c] | ≤0.001 |
| Emotional/mental health subscale (Cronbach α = 0.71) | 3.10±0.84 | 2.35 ±0.81[a] | 3.01 ± 0.74[b] | 3.70±0.79[c] | ≤0.001 |
| Awareness subscale (Cronbach α = 0.90) | 2.84±1.00 | 2.39 ±0.71[a] | 2.66 ± 0.91[a] | 3.64±0.99[b] | ≤0.001 |
| Total (Cronbach α = 0.90) | 2.94±0.75 | 2.39 ±0.59[a] | 2.81 ± 0.64[b] | 3.60±0.73[c] | ≤0.001 |
| **Perceived Benefits** | | | | | |
| Emotional/mental health subscale (Cronbach α = 0.85) | 3.89±0.67 | 4.04 ±0.46[a] | 3.92 ±0.63[a] | 3.51 ±0.94[b] | 0.002 |
| Physical health/fitness subscale (Cronbach α = 0.90) | 3.80±0.65 | 3.87 ± 0.71 | 3.79 ±0.64 | 3.47 ±0.99 | 0.093 |
| Social/professional subscale (Cronbach α = 0 .75) | 3.54±0.85 | 3.50 ±0.84 | 3.57 ±0.84 | 3.32 ±1.01 | 0.171 |
| Total (Cronbach α = 0.92) | 3.73±0.67 | 3.87 ± 0.58 | 3.80 ±0.57 | 3.46 ±0.72 | 0.044 |
| **Cue to action** | | | | | |
| Internal cues (Cronbach α = 0.85) | 3.57±0.76 | 3.61 ± 0.58 | 3.62 ± 0.70 | 3.40 ±0.96 | 0.512 |
| External cues (Cronbach α = 0.86) | 3.41±0.77 | 3.45 ± 0.54 | 3.47 ± 0.72 | 3.21 ±0.97 | 0.121 |
| Total (Cronbach α = 0.90) | 3.49±0.70 | 3.53± 0.49 | 3.54± 0.65 | 3.30± 0.90 | 0.228 |
| **Perceived self-efficacy in dieting** | | | | | |
| Habits and preferences subscale (Cronbach α = 0.84) | 3.24±0.66 | 3.72±0.52[a] | 3.28±0.62[b] | 2.90±0.69[c] | ≤0.001 |
| Emotional/mental health subscale (Cronbach α = 0.84) | 3.20±0.96 | 3.99±0.43[a] | 3.27±0.96[b] | 2.66±0.86[c] | ≤0.001 |
| Total (Cronbach α = 0.88) | 3.22±0.64 | 3.81±0.42[a] | 3.27±0.58[b] | 2.82±0.67[c] | ≤0.001 |
| **Perceived self-efficacy in exercise** | | | | | |
| Total (Cronbach α = 0.83) | 3.27±0.79 | 3.23±0.75[ab] | 3.39 ± 0.71[a] | 2.90 ± 0.95[b] | 0.001 |
| **Behavioral intention of weight management** | | | | | |
| Diet therapy subscale (Cronbach α = 0.77) | 2.93±0.95 | 3.21±0.66 | 2.86±0.95 | 3.05±1.03 | 0.096 |
| Exercise therapy subscale (Cronbach α = 0.72) | 3.28±0.94 | 3.00±0.95 | 3.31±0.93 | 3.31±0.96 | 0.299 |
| Total (Cronbach α = 0.77) | 3.07±0.78 | 3.13±0.54 | 3.04±0.77 | 3.15±0.90 | 0.544 |

P-values are based on Kruskal-Wallis Test.

[a, b, c] Values within a row with the same letter indicate no significant difference. Any difference between two values carrying different letters is significant at 0.05 based on Mann—Whitney U with Bonferroni Correction.

The mean rating on the self-efficacy in dieting was 3.22±0.64 for all respondents that showed significant differences among three groups (P≤0.001). As, students in the underweight group showed the strongest belief about their self-efficacy in dieting (3.81±0.42); followed by students in the normal-weight (3.27±0.58) and overweight group (2.82±0.67).

The mean rating on the self-efficacy in exercise was 3.27±0.79 for all respondents. Students in the normal-weight group had the highest score (3.39±0.71) and indicated significant differences in comparison to those in the overweight group (P≤0.001). But these two groups showed no significant difference, compared to the underweight group.

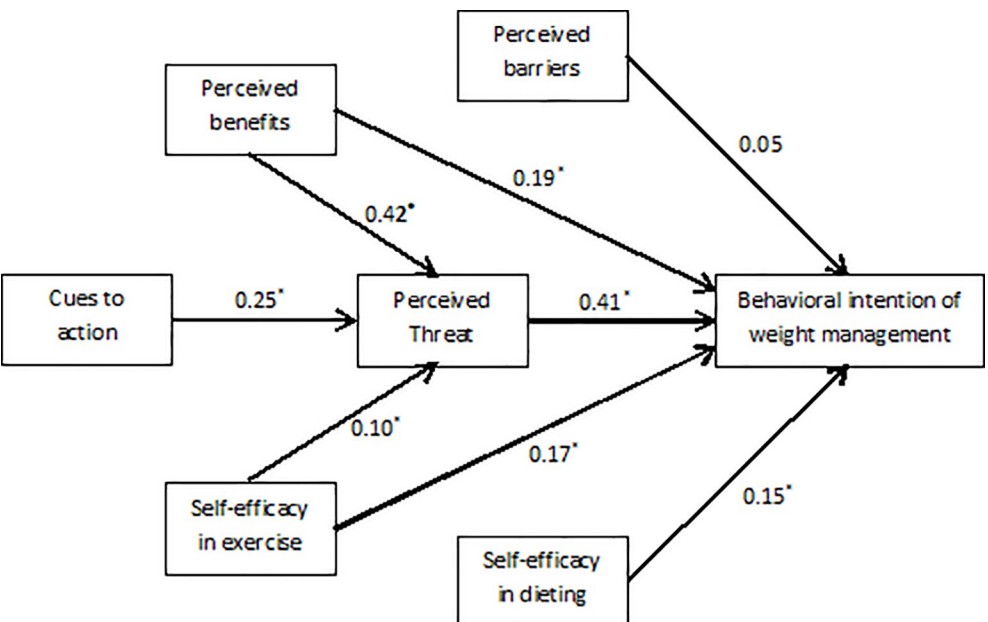

**Fig 2. Effects of Health Belief Model constructs on behavioral intention of weight management.** Path coefficients were shown above. *Significant at 0.05 level. χ2/df = 2.68, CFI = 0.99, TLI = 0.95, RMSEA = 0.07, SRMR = 0.02.

The mean rating on behavioral intention of weight management was 3.07±0.78 for all respondents. The result showed that students intended to manage their weight by exercising rather than dieting. The mean score of the total behavioral intention of weight management and the two subscales did not demonstrate significant differences among the three groups.

## Path models

Effects of the final model of HBM constructs on weight management behaviors are displayed in Fig 2. This model was identified given the good fit indices (χ2/df = 2.68, CFI = 0.99, TLI = 0.95, RMSEA = 0.07, SRMR = 0.02) for the all students sample. The model indicated that perceived threats, perceived barriers, perceived benefits, self-efficacy in dieting and self-efficacy in exercise directly affected behavioral intention of weight management. Higher level of aforementioned scales was significantly related to greater behavioral intention of weight management. Moreover, cues to action, perceived benefits and self-efficacy in exercise indirectly affected behavioral intention of weight management through the impact of perceived threats. Tables 3 shows total, direct, and indirect effects of HBM constructs on weight management behavior. Perceived threats and perceived benefits were the greatest predictor of weight loss behaviors with a total correlation coefficient of 0.40 and 0.35, respectively. All of these associations were significant, except for the association of perceived barriers and behavioral intention of weight management.

## HBM as a predictor for BMI

Table 4 presents findings from the two-step hierarchical regression analysis constructed to test the ability of HBM and some of the general characteristics to predict the BMIs of college students. The models were constructed from data provided by all students who responded to the whole HBM scale. When perceived severity, perceived susceptibility, perceived benefits, perceived barriers, cues to action, and self-efficacy in dieting and self-efficacy in exercise were

**Table 3. The total, direct and indirect effect of Health Belief constructs on behavioral intention of weight management.**

|  | Direct | Indirect | Total |
|---|---|---|---|
| Perceived threat | .41* | - | 0.41* |
| Perceived benefits | 0.18* | 0.17* | 0.35* |
| Perceived barriers | 0.06 | - | 0.05 |
| Cues to action | - | .10* | 0.10* |
| Self-efficacy in dieting | 0.15* | - | 0.15* |
| Self-efficacy in exercise | 0.17* | 0.04* | 0.21* |

*Significant at 0.05 level.

regressed against BMI, the model was highly significant (P<0.001). The first model explained approximately 34% of the variance of the students' BMIs. Self-efficacy in dieting and perceived severity had an inverse significant association with BMI. Self-efficacy in dieting (β = -1.63, P<0.001), perceived barriers (β = 1.18, P<0.001), and perceived severity (β = -1.17, P<0.001) seemed to be the most important among these seven variables. Findings also revealed significant positive associations between ratings on the perceived barriers and BMI. In model two, those demographic variables that had a significant correlation with BMI were added to model 1. The inclusion of age and marital status increased the $R^2$, and explained 40% of the variance in BMI (P<0.001).

## Discussion

The present study was conducted to investigate the factors influencing behavioral intention by applying HBM and estimate the relationships between several belief scales and the BMIs of students. Weight loss is usually less successful, despite applying various weight-loss programs, available to the public; once succeeded, the maintenance as well as long-term weight-loss program compliance rates are usually low [30]. Therefore, the identification of psychological predictors of weight management could contribute to improv treatment efficacy [15–17].

The present results showed that students with different weight statuses had different perceptions about obesity and weight reduction behavior. The constructed SEM in this study

**Table 4. Hierarchical multiple regression analysis for predicting body mass index.**

| Independent Variable | Model 1 | | | | Model 2 | | | |
|---|---|---|---|---|---|---|---|---|
|  | B | SE | Beta | P- value | B | SE | Beta | P- value |
| Perceived severity | -1.08 | 0.25 | -0.26 | <0.001 | -1.17 | 0.24 | -0.28 | <0.001 |
| Perceived susceptibility | 0.38 | 0.23 | 0.09 | 0.083 | 0.40 | 0.22 | 0.10 | 0.064 |
| Perceived benefits | 0.06 | 0.31 | 0.01 | 0.849 | 0.08 | 0.30 | 0.01 | 0.788 |
| Perceived barriers | 1.30 | 0.23 | 0.31 | <0.001 | 1.18 | 0.22 | 0.28 | <0.001 |
| Cues to action | 0.37 | 0.27 | 0.08 | 0.175 | 0.46 | 0.26 | 0.10 | 0.078 |
| Self-efficacy in dieting | -1.50 | 0.26 | -0.30 | <0.001 | -1.63 | 0.26 | -0.33 | <0.001 |
| Self-efficacy in exercise | 0.21 | 0.21 | 0.05 | 0.336 | 0.31 | 0.21 | 0.08 | 0.129 |
| Age |  |  |  |  | 0.23 | 0.05 | 0.22 | <0.001 |
| Marital status* |  |  |  |  | -0.97 | 0.48 | -0.09 | 0.042 |
| Adjusted $R^2$ | 0.34 |  |  |  | 0.40 |  |  |  |

*Reference group was those married.

Dependent variable was BMI.

supported the theoretical framework, indicating that health beliefs can directly and indirectly predict student's behavior intention for weight management. In addition, the HBM scales partially predicted the students' BMIs.

The current finding showed that the most common weight management methods among students were exercise and dieting. This result is consistent with those of other studies that examined weight-loss practices among university students [31, 32]. Nearly, 55% of the students responded that they attempted to control their weight for a better appearance. The current findings are in line with those of other studies which have indicated that keeping up appearance was the main reason for managing body weight among university students [31]. The socioeconomic conditions of the participants were not related to their weight status. Previous studies have reported contradictory results regarding the association between socioeconomic status and BMI [20, 33–35]. The lack of standard methods for categorizing SES might be the main reason for this contradiction [36].

Overweight students in comparison with other groups showed lower ratings on perceived severity and self-efficacy in dieting and self-efficacy in exercise, but higher ratings on all subscales of perceived barriers to adopting healthy eating and physical activity habits. The higher ratings on the severity belief scale given by underweight and normal-weight students may have motivated them to manage their weight, since individuals make changes if they perceive that their current status could have serious health complications. Some previous studies have shown that obese people have less perceived self-efficacy in relation to eating and exercise than non-obese groups [37–39]. Participants' perceived self-efficacy reflects the confidence in their capacity to perform a new health behavior. A person with a higher level of confidence will more likely engaged in a specific healthy eating behavior to improve health. In this regard, it has been reported that obese Americans are more likely to name several barriers to weight-loss behaviors, compared with non-obese individuals [40]. The results demonstrated that emotional/mental factors, unawareness of healthy food choices, and practical obstacles hamper students to refrain from unhealthy eating behaviors or calorie-dense foods. Moreover, underweight and normal-weight students gave higher, but not significant ratings to perceived susceptibility beliefs than overweight students. Unlike previous studies, the current results suggested that these groups of students may not consider themselves susceptible enough to being overweight to take further action. Moore *et al*. reported that African American normal-weight women reported the same perceived threat of obesity-related diseases as overweight women [41]. In fact, an inappropriate perception of one's own weight and inadequate information about the consequences of obesity could lessen the perceived threat of being obese. Students in underweight and normal-weight groups showed the strongest beliefs about the emotional/mental benefits of adopting healthy eating and physical activity habits. Differences did not reach the significance level in other subscales of perceived benefit. These results are inconsistent with prior research [42, 43]. Such findings suggest that anticipation of the favorable outcomes of adopting healthy eating habits and engaging in regular physical activity can encourage participants to manage their weight.

In the present study, the constructed SEM provides a better understanding of the mechanism through which psychosocial factors affect weight management behavior. The results of path analysis indicated that perceived variables including threat and self-efficacy in dieting, have a significant direct effect and perceived benefits and self-efficacy in exercise have significant direct and indirect effects on predicting weight management behavior. Higher levels of the mentioned perceptions further predicted a higher chance of executing behavioral intention of weight management. Perceived threat exerted the greatest influence on behavioral intention of weight management in all respondents, followed by perceived benefit. These results are in agreement with those that suggest perceived benefits, threat, and self-efficacy as strong

predictors of some health behaviors [42–44]. Bishop *et al*. reported that perception of threat and self-efficacy account for a considerable amount of the variance in the performance of patient safety practices [44]. When the rate of self-efficacy or person's confidence in their ability to perform a specific behavior was high, the probability of incorporating health behavior changes was also increased. O'Connell *et al*. found that dieting benefits were the most powerful predictor of dieting behavior, especially for obese adolescents [43]. In a study by Kang *et al*., perceived benefits was the most important predictor of intention to control obesity among female students [42]. This result suggests that if patients are aware of the benefits of managing weight by dieting and exercise, they might become involved in the programs.

The results showed that perceived barriers to eating healthy foods and to undertaking regular physical activity could not significantly affect behavior intention of weight management. This result was consistent with the results of some [20, 45], but not other [46, 47] studies which have reported that a higher perception of the difficulties and cost of performing behaviors are negatively related to a lower likelihood of performing the positive health behaviors. In the present study, the perceived barriers were increased among students living in dormitories due to problems such as lack of time, insufficient knowledge, and insufficient skills in preparing healthy food [48, 49]; thus this component failed to justify the behavioral intention of weight management.

In the present research, perceived threat mediated the relationship between cues to action and behavioral intention of weight management. This suggests that external and internal cues would arouse a person's perceived threat of the risk of obesity by influencing perceived seriousness, susceptibility, or both which led the students to weight management behavior. For example, the person believes that others judge her unfairly, owing to her weight or an obese family member or a friend, and a serious health problem developed from being obese.

In both regression analysis models, perceived severity, perceived barriers, and self-efficacy in dieting were the significant variables in predicting the BMIs of all respondents. Self-efficacy in dieting seemed to be the most significant parameter among the three variables. The final model, in which the demographic variables were added, explained approximately 40% of the variance of students' BMIs. The results of the current study showed that students who assumed themselves to be confident in their ability to perform the behavior had lower BMIs. This is compatible with previous results showing that obese women scored significantly less than the non-obese on self-efficacy in relation to food [37]. The significant inverse association between perceived severity and students' BMIs in both regression models proposed that students who noticed the possible negative physiological, psychological, and social consequences of being obese (*e. g*., chronic disease, mental health problems, difficulties in social relationship) had lower BMIs. The significant positive associations between the ratings of the perceived barriers scales and students' BMIs suggested that participants who regarded difficulties (*e. g*., lack of time, knowledge, and motivation) and cost of performing behaviors had higher BMIs.

There were several worth noting limitations in the design and performance of this study. The main limitation was the cross-sectional, non-experimental design, which provides only a glimpse of the population at a specific point of time. In addition, only dormitory students of medical sciences were included herein, which confines the generalizability of the findings to all college students. Moreover, all the subjects were females, that are more prone to control eating habits and weight. Also, the anthropometric data was collected through self-reporting and data was collected through personal interviews that could lead to bias in the results. Future studies are needed to use HBM to identify the associations between weight-related beliefs of diverse samples and their weight management behaviors. In addition, it would be worthwhile to expand interventional studies to investigate the effect of HBM-based educational programs on weight management in college students or other populations.

## Conclusions

The significant variables in predicting behavioral intention of weight management were perceived threat, perceived benefits, self-efficacy in dieting and self-efficacy in exercise, and cues to action. In addition, it was reported that students have different perceptions about obesity and weight reduction behavior by weight status. These results suggest that to ensure the adherence and success of the participants in health intervention, it is essential to design and implement health education programs along with dietary approaches. Such programs should emphasize the negative outcomes of obesity, benefits of adopting a healthy lifestyle, increase of self-efficacy in dieting and physical activity, and internal and external stimuli for college students.

## Supporting information

**S1 File. Questionnaire in English.**
(PDF)

**S2 File. Questionnaire in Persian.**
(PDF)

## Acknowledgments

The authors would like to thank Dr. Hossein Karimzadeh, Assistant Professor, Department of Urban Planning of Tabriz University for his assistance with data analysis, and also the participants who involved in this study. Also, the authors would like to acknowledge the support of this work by Student Research Committee of Tabriz University of Medical Sciences.

## Author Contributions

**Conceptualization:** Maryam Saghafi-Asl, Soghra Aliasgharzadeh, Mohammad Asghari-Jafarabadi.

**Data curation:** Maryam Saghafi-Asl, Soghra Aliasgharzadeh.

**Formal analysis:** Soghra Aliasgharzadeh, Mohammad Asghari-Jafarabadi.

**Investigation:** Maryam Saghafi-Asl.

**Methodology:** Mohammad Asghari-Jafarabadi.

**Project administration:** Maryam Saghafi-Asl, Soghra Aliasgharzadeh.

**Resources:** Soghra Aliasgharzadeh.

**Software:** Mohammad Asghari-Jafarabadi.

**Supervision:** Maryam Saghafi-Asl.

**Validation:** Mohammad Asghari-Jafarabadi.

**Writing – original draft:** Soghra Aliasgharzadeh.

**Writing – review & editing:** Maryam Saghafi-Asl, Mohammad Asghari-Jafarabadi.

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
