## [Decision Letter · Decision Letter 0]

10 Oct 2019

PONE-D-19-22092

Factors influencing weight management behavior among College Students: An application of the Health Belief Model

PLOS ONE

Dear Dr. Aliasghazadeh

Thank you for submitting your manuscript to PLOS ONE. After careful consideration, we feel that it has merit but does not fully meet PLOS ONE’s publication criteria as it currently stands. Therefore, we invite you to submit a revised version of the manuscript that addresses the points raised during the review process.

In this regard, you should review each suggestion made by the reviewers, especially those associated with the limitations of the study and update the references. In addition, you should establish each hypothesis in the introduction and contrast them in the results section (Figures 1 and 2), including the mediation relationships found.

We would appreciate receiving your revised manuscript by November 26. To enhance the reproducibility of your results, we recommend that if applicable you deposit your laboratory protocols in protocols.io, where a protocol can be assigned its own identifier (DOI) such that it can be cited independently in the future. For instructions see: http://journals.plos.org/plosone/s/submission-guidelines#loc-laboratory-protocols

We look forward to receiving your revised manuscript.

Kind regards,

Berta Schnettler

Academic Editor

PLOS ONE

Journal Requirements:

1. Please include additional information regarding the survey or questionnaire used in the study and ensure that you have provided sufficient details that others could replicate the analyses. For instance, if you developed a questionnaire as part of this study and it is not under a copyright more restrictive than CC-BY, please include a copy, in both the original language and English, as Supporting Information. Additionally, clarification of the sampling method and reason for chosen sample size would be beneficial.

2.

We suggest you thoroughly copyedit your manuscript for language usage, spelling, and grammar. If you do not know anyone who can help you do this, you may wish to consider employing a professional scientific editing service.  

Reviewers' comments:

Reviewer's Responses to Questions

**Comments to the Author**

1. Is the manuscript technically sound, and do the data support the conclusions?

Reviewer #1: Yes

Reviewer #2: Yes

2. Has the statistical analysis been performed appropriately and rigorously? 

Reviewer #1: Yes

Reviewer #2: Yes

3. Have the authors made all data underlying the findings in their manuscript fully available?

Reviewer #1: Yes

Reviewer #2: Yes

4. Is the manuscript presented in an intelligible fashion and written in standard English?

Reviewer #1: No

Reviewer #2: Yes

5. Review Comments to the Author

Reviewer #1: Thank you for asking me to review this intersting study on factors influencing weight management in university students.

Here my cooments for the authors:

1. At the end of the abstract, the authors used the word "prosperity", maybe they would state health or wellbeing

2. About the methods, can the authors explain why a sample of 336 female subjects was considered representative?

I personally appreciate if the author would consider to report the questionnaire with all items to help other researchers to replicate the study

3. Table 1: I suggest to better explain the meaning of letter a-b- c

4. Table 2 is quite difficult to read. I suggest to improve the readability separating it in different sections

5.In Results section: Baseline characteristics in table students experiencin weight loss behaviour are 146/336 (43%) in the text they are more than half

6. Analysing table 3, the authors mentioned in the text bahavioural intention of weight management, not present in table 3

7. In Discussion section, the authors mentioned higher ratings on the severity belief scale may be due to factors like lack of time, insufficient knowledge and skill in food preparations. These factors are maybe due to the fact all the subjects involved in the study are living in a campus. Several studies pointed out worsening behaviours in eating habits of university students leaving family and difference between students living alone or with family (see for instance and comment PMID 17368642 and 26156187)

8. In the limitations of the study, I suggest to include also that all subjects were females, that are, as everyone knows, more prone to control eating habits and weight, in addition data were collected by personal interviews and that could affect the response

9. There are some typing errors (i.e. wit it instead of with it in introduction section). I recommend a revision of the text

Reviewer #2: The study seems relevant and provides scientific evidence based on theoretical framework. However, It is important to point out that the results of this study only pertain to the population that was studied and cannot be extrapolated to other populations. In addition, more socio-demographic data of the participants is needed. Please review the comments included in the attached file.

6. PLOS authors have the option to publish the peer review history of their article (what does this mean?). If published, this will include your full peer review and any attached files.

Reviewer #1: No

Reviewer #2: Yes: Luis Horacio Aguiar Palacios

---

## [Author Response · Author response to Decision Letter 0]

16 Dec 2019

Dear Dr. Berta Schnettler,

Respected editor, "PLOS ONE",

Thank you so much for your email dated 2019. The reviewers’ comments were extremely helpful, and we believe that the revisions have enhanced the quality of the manuscript substantially. We have addressed all the comments of the reviewers. Please kindly notice that one-by-one responses to the reviewers’ comments is provided in a “Response to Reviewers” and all the new changes and inclusions in the main body of manuscript in a “Revised Manuscript with Track Changes” Word file. If you have further question please let us know as soon as possible.

We sincerely hope that the revised manuscript has been efficiently improved and reached to the level of acceptance in the PLOS ONE.

Best respects,

Soghra Aliasgharzadeh

Our response to each comment is as below:

Editor’ comments:

1. Regarding the editor comments “you should establish each hypothesis in the introduction and contrast them in the results section (Figures 1 and 2), including the mediation relationships found.”

RESPONSE: Each hypothesis in the introduction section related to effects of HBM constructs on behavioral intention of weight management (figure 1) and the corresponding results in the related section was described; however, since it would be so dense the related results was only shown in one paragraph.

2. Please include additional information regarding the survey or questionnaire used in the study and ensure that you have provided sufficient details that others could replicate the analyses. For instance, if you developed a questionnaire as part of this study and it is not under a copyright more restrictive than CC-BY, please include a copy, in both the original language and English, as Supporting Information. Additionally, clarification of the sampling method and reason for chosen sample size would be beneficial.

RESPONSE: Your comment is perfectly reasonable. Additional information regarding the sample size estimation and sampling method were added in the method section. A copy of questionnaire in both the original language and English were attached as supporting information.

3. Regarding the English editing of the manuscript:

RESPONSE: The manuscript was edited for language and grammars by professional scientific editing service and the EDITORIALCERTIFICATELETTER has been attached.

Reviewer 1:

Thank you so much for comments:

1. At the end of the abstract, the authors used the word "prosperity", maybe they would state health or wellbeing

RESPONSE: The word “prosperity” was replaced by “well-being” in the abstract.

2. About the methods, can the authors explain why a sample of 336 female subjects was considered representative?

I personally appreciate if the author would consider to report the questionnaire with all items to help other researchers to replicate the study

RESPONSE: With regard to this comment, additional information about sample size estimation was added in the method section. The questionnaire used in the study were attached as supporting information.

3. Table 1: I suggest to better explain the meaning of letter a-b- c

RESPONSE: Better explanation about the a, b, c letters was added to tables caption.

4. Table 2 is quite difficult to read. I suggest to improve the readability separating it in different sections

RESPONSE: We tried to improve the readability of the table 2 by changing the color of heading each section.

5. In Results section: Baseline characteristics in table students experience in weight loss behaviour are 146/336 (43%) in the text they are more than half

RESPONSE: The phrase “more than half” were corrected in to “Nearly half” in the mentioned sentence.

6. Analysing table 3, the authors mentioned in the text bahavioural intention of weight management, not present in table 3

RESPONSE: Behavioral intention of weight management is a dependent variable and caused and influenced by listed variables.

7. In Discussion section, the authors mentioned higher ratings on the severity belief scale may be due to factors like lack of time, insufficient knowledge and skill in food preparations. These factors are maybe due to the fact all the subjects involved in the study are living in a campus. Several studies pointed out worsening behaviours in eating habits of university students leaving family and difference between students living alone or with family (see for instance and comment PMID 17368642 and 26156187)

RESPONSE: The mentioned references were added to the discussion section where the higher ratings on the barriers scale may be due to factors like lack …

8. In the limitations of the study, I suggest to include also that all subjects were females, that are, as everyone knows, more prone to control eating habits and weight, in addition data were collected by personal interviews and that could affect the response

RESPONSE: Suggestions about limitations were applied.

9. There are some typing errors (i.e. wit it instead of with it in introduction section). I recommend a revision of the text

RESPONSE: The typing errors were corrected.

Reviewer 2:

Thank you so much for comments:

1. The study seems relevant and provides scientific evidence based on theoretical framework. However, it is important to point out that the results of this study only pertain to the population that was studied and cannot be extrapolated to other populations.

RESPONSE: Explanation about the generalizability of the results considering the studied population was added and highlighted.

2. There is a lack of information about participants socio-demographic characteristics, like age or nationality.

RESPONSE: With regard to this comment, information about nationality of participants added to methods section. As far as we could, we presented all the socio-demographic data of the participants; however, unfortunately we do not have access to the same population now to inquire more information.

3. Please rename this table. Mean and Cronbach Alpha must be specified at the end of the table.

RESPONSE: Table 2 was renamed according to the reviewer’s comment.

4. In discussion section, a statement is part of theoretical framework.

RESPONSE: That sentence was deleted

5. Only 32% of the references are at least five years old, 23% are about 10 years old which is ok but 45% are too old. It is convenient to have more references from recent years.

RESPONSE: Some references were replaced with recent literature (references from recent years), as much as possible. However, some could not be replaced with new ones, as they were only done in old years. Since such references are usually related to hypotheses which have been established several years ago and therefore, they could not be substituted with recent ones.

---

## [Decision Letter · Decision Letter 1]

7 Jan 2020

Factors influencing weight management behavior among College Students: An application of the Health Belief Model

PONE-D-19-22092R1

Dear Dr. Aliasgharzadeh

We are pleased to inform you that your manuscript has been judged scientifically suitable for publication and will be formally accepted for publication once it complies with all outstanding technical requirements.

With kind regards,

Berta Schnettler

Academic Editor

PLOS ONE

Additional Editor Comments (optional):

Reviewers' comments:

Reviewer's Responses to Questions

**Comments to the Author**

1. If the authors have adequately addressed your comments raised in a previous round of review and you feel that this manuscript is now acceptable for publication, you may indicate that here to bypass the “Comments to the Author” section, enter your conflict of interest statement in the “Confidential to Editor” section, and submit your "Accept" recommendation.

Reviewer #1: All comments have been addressed

Reviewer #2: All comments have been addressed

2. Is the manuscript technically sound, and do the data support the conclusions?

Reviewer #1: (No Response)

Reviewer #2: Yes

3. Has the statistical analysis been performed appropriately and rigorously? 

Reviewer #1: (No Response)

Reviewer #2: Yes

4. Have the authors made all data underlying the findings in their manuscript fully available?

Reviewer #1: (No Response)

Reviewer #2: Yes

5. Is the manuscript presented in an intelligible fashion and written in standard English?

Reviewer #1: (No Response)

Reviewer #2: Yes

6. Review Comments to the Author

Reviewer #1: (No Response)

Reviewer #2: The latest version was considered approved. We do have to point out that this does not mean a definitive approval from the magazine. The criteria of the other reviewers should be taken into consideration.

After a careful and exhausting revision it has been decided to approve the article titled “Factors influencing weight management behavior among College Students: An application of the Health Belief Model” in the hope that the criteria mentioned would help the author improve on his future work and thus the magazine keep publishing articles with the highest of quality. Best regards.

7. PLOS authors have the option to publish the peer review history of their article (what does this mean?). If published, this will include your full peer review and any attached files.

Reviewer #1: No

Reviewer #2: Yes: Luis Horacio Aguiar Palacios

---

## [Editor Report · Acceptance letter]

31 Jan 2020

PONE-D-19-22092R1 

Factors influencing weight management behavior among College Students: An application of the Health Belief Model 

Dear Dr. Aliasgharzadeh:

I am pleased to inform you that your manuscript has been deemed suitable for publication in PLOS ONE. Congratulations! Your manuscript is now with our production department. 

With kind regards,

on behalf of

Dr. Berta Schnettler 

Academic Editor

PLOS ONE